# Incarcerated Brazilian Elderly: Memories about Family

Pollyanna Lima [1], Alessandra Oliveira [1], Luana Reis [1], Arianna Lopes [1], Elaine Santana [2], Thaiza Nobre [3] and Luciana Reis [4],*

[1] Independent School of the Northeast, Collegiate University of Nursing, Vitória da Conquista 45055-030, Brazil
[2] Health Sciences Research Unit, Nursing School of Coimbra (ESEnfC), 3000-232 Coimbra, Portugal
[3] Department of Physiotherapy, Federal University of Rio Grande do Norte, Natal 59200-000, Brazil
[4] Department of Health I, State University of Southwest Bahia, Jequié 45208-177, Brazil
* Correspondence: luciana.araujo@uesb.edu.br

**Abstract:** This article aims to analyze the memories of incarcerated elderly people about family. This is an exploratory and analytical study, with a qualitative approach, carried out in three prison units in Bahia, with 31 incarcerated elderly people, through semi-structured interview. Most are male (30), between 60 to 65-years-old (21), married (12) and with three to four children (10). The results show that the 10 most evoked words were: family; sons; mom; father; women; life; cry; today; brothers; and longing. The study showed that the family constitution is maintained because the experiences that are symbolized in it have socially crystallized definitions as references. Anchored by social frames of memories that remain, since they remain alive in the groups and are permanently maintained. It was also found that the family plays an extremely important role in their lives and that the mother is the central point of this family.

**Keywords:** family; seniors; prison; memory





## 1. Introduction

In 2017, the elderly population in Brazil passed the mark of 30.2 million according to the Continuous National Household Sample Survey [1], and, in accordance with the projections of The Brazilian Institute of Geography and Statistics (IBGE), the Brazilian elderly population must reach 58.2 million, that is to say, 25.5% of the total population, by 2060 [2].

The demographic trend of the elderly population growth in the country is also noticed in the prison environment, with the rise in crime cases perpetrated by elderly people in Brazil and in the world, but the focus of this study is the Brazilian case. In accordance with data from the National Penitentiary Department [3], in the period between December 2008 and December 2011, there was a rise of 45.91% in the imprisoned elderly population. The amount of imprisoned elderly in the Brazilian penal system in 2012 was 5045 people, of which 4771 were males and 274 females, which corresponds to 0.92% of the 548,003 people imprisoned in the country. In June of 2013, the amount of imprisoned elderly in the Brazilian penal system was 5333 people: 5012 were males, and 321 females. These data show the growth in the number of elderly people imprisoned in Brazil [3].

The prison environment is insalubrious and affects different aspects of the elderly's life, such as the biological, economic, social and psychological. The prison situation as a whole strengthens physical and mental symptoms, since the increased number of prisoners with health problems living in the same environment, in very crowded cells, brings negative consequences to the physical and mental conditions [4].

The elderly subject, when experiencing prison, faces, besides physical and mental problems, the distancing of their social relations with the world on the other side of the prison walls. The interaction with friends, family and society becomes practically non-existent, restricting themselves to the family visits in prison, when they are not forgotten by them.

Research points out that subjects who do not have family support present a worse confrontation in the face of adversity, since family support constitutes a positive impact on the psychological and emotional well-being of the elderly people [5,6].

For many elderly people, the family is linked to the perception of cohesion and emotional comfort that comes from their relationships with people that are meaningful to them. The family's presence in times of crisis and particular need may help to face stressful events such as prison and its effects, contributing to the psychological well-being of the elderly people and then positively influencing their physical and mental health [7].

Away from the interaction with their relatives, the memory of this relation and the role that he/she had in the family remains with the imprisoned elderly person. A study by Halbwachs [8] regarding collective memory verified that, whatever the memories of the past a subject has, and as much they seem to be a result of exclusively personal experiences, memories of the family come from the memory's social frameworks.

The memory's social frames are "instruments that the collective memory uses to rebuild an image of the past according to each period and in line with society's dominant thoughts" [8]. In accordance with the author's assumptions, no memory is possible outside the social frames in which the subjects are inserted.

Halbwachs [8] sets as the family memory's framework "faces and achievements that set themselves up as points of reference, but each one of these faces indicates a personality, each one of these achievements summarize an entire period of the group life".

By understanding the significance of family relations for the conviviality of the elderly individuals, in this article, we seek to deepen the memories they have of their families and how these memories interfere with their behavior in prison. In this sense, in order to support future strategies for a greater insertion of the family in the prison context, this study aims to analyze incarcerated elders' memories about family in order to contribute to future effective public health interventions aimed at raising awareness and motivation to increase the support network for incarcerated elders.

Thus, this study initially presents the methodological approach, then the results and discussion, based on the interpretation of the reports of the Brazilian incarcerated elderly, but also in view of the literature that portrays the prison and the family institution.

## 2. Materials and Methods

### 2.1. Study Design

It is an exploratory and descriptive study, with a qualitative method, and the Collective Memory is the theoretical-methodological input. It is a excerpt of the thesis named "Memory and Representations of arrested elderly people on Aging and Health", which has as Presentation Certificate for Ethical Assessment (CAAE) no. 65550217.8.0000.5578 and approval opinion no. 1.968.281. This is a self-funded study.

In order to meet the proposed objectives, this study was characterized as exploratory and descriptive, with a qualitative approach, and has the Collective/Social Memory as its theoretical and methodological contribution. Memory was taken as an object and analytical resource, which became a hermeneutic resource, that is, it became an instrument of interpretation (MONTESPERELLI, 2004). Through the speeches produced by the incarcerated elderly, it was possible to mobilize their memories and thus achieve the goal of this study.

### 2.2. Sample and Data Collection Procedure

The study was performed in three Prison Units (PUs) in the interior of Bahia, currently tied to the State Secretariat of Penitentiary Administration and Resocialization (SEAP), according to the Law No. 12,212 of 4 May 2011 [9]. Bahia's prison population is 14,916, including the female and the male genders, and of these, 1639 are in the PUs where the study was carried out [10]. The choice of the PUs in Bahia was due to the convenience of being closer to the researchers, as well as the portrayal of very similar realities.

The number of elderly prisoners in the three PUs varied a lot during the months in which the collection took place, because the prisoners' rotation is very intense. Besides the

freedom of some, some were transferred, others died, and new ones arrived; therefore, the number of elderly prisoners varied weekly. As a result, the participants of the present study were 31 people aged 60-years-old or more who were able to participate in the research because they were in prison, placed in one of the PUs selected for the research, and whose cognitive functions were preserved, assessed by the Mini Mental State Examination.

The complete MMSE is comprised of two sections, which assess cognitive functions. The first section measures orientation, memory and attention, totaling 21 points. The second part measures the ability to follow a verbal and written command and to copy a complex drawing of a polygon, totaling nine points. The total score is 30 points, and the cutoff point is 23/24, which is a score that suggests a cognitive deficit [11]. This mini mental state examination enabled the exclusion of elderly subjects who did not have enough cognition ability to participate in the study.

A semi-structured interview script was used, with the topic of family, with the intention to obtain the memories of the imprisoned elderly.

The field research took place from March to October 2017, totaling seven months of collection, always at times previously scheduled with the PUs. After authorization for the field research and contact with the directors, we personally delivered the field release and infrastructure authorization documents signed by the SSP-BA, as well as the opinion of the CEP in the prison units. Moreover, a schedule for data collection was built based on the availability of the unit itself, since the PUs had particularity different schedules from one another.

The first contact also made it possible to get to know the structure of the facilities, the way they operate, the schedules, the rules, and to have contact with some prisoners who work during the day at the facilities. Furthermore, in this first meeting, a filtering of the research participants was carried out, according to the inclusion and exclusion criteria already described. After these procedures, the interview process was then started.

*2.3. Data Analysis*

The semi-structured interview was examined through the Content Analysis technique described by Laurence Bardin, in the modality of thematic analysis [12], with the assistance of the NVivo version 11 computing tool to support the qualitative data analysis [13]. Content analysis [12] encompasses a sequence of analysis techniques that uses systematic procedures, organized in three stages: pre-analysis, exploring the material and the treatment of outcomes. Afterwards, the results were addressed with inference and interpretation and with the articulation of the empirical work and the theoretical reference of memory.

To represent the outcomes in a panoramic way and to enable the visualization, the NVivo resource named "Words Cloud" was used. To build the word cloud, the frequency of the words was determined with the categories in the NVivo software, and the option to generate the words cloud was used. At that moment, the researcher could choose the words exclusion, when she thought that some words were irrelevant to the study or that they did not have semantic meaning, such as in the case of prepositions. For that, the software option was used to prevent the word from being added to the content analysis process. Apart from this possibility of removing undesired words, it was also possible to choose the number of words for each cloud displayed. In this study, we chose to present only the 50 most mentioned words, as we believed that the cloud would be more presentable and the research's corpus would be representative with more evidence.

Bearing in mind that it is a study with human beings, and in a place with many particularities, it is highlighted that it was an incessant effort to fulfill all the ethical and scientific foundations in a rigorous manner. The participants were respected in their dignity, weighing up the risks and benefits from the time when the study was considered. This way, the guarantees that damages would be prevented were made, to perform a study in which social relevance was the premise.

*2.4. Ethical Issues*

All participants were aware of the study and its goals and methodology and signed the informed consent form according to the resolutions of The National Health Council (CNS) no. 466/2012 and 510/2016 for human research [14,15]. Instead of the elderly people's names, with the purpose of maintaining anonymity, the letters I (Imprisoned), E (Elderly) and the numerical sequence 01 to 31 were used, since there were 31 study participants for example, the first participant was named as: (IE-01).

**3. Results**

The 31 elderly participants of the study are mostly male (30), aged from 60 to 65-years-old (21), brown-skinned/"pardo" (16), with no schooling degree (12), married (12), with three to four children (10), of the evangelical religion (17), predominantly worked as farmworkers (14), had an urban origin (16) and had prison time from one to two years (16).

The family remains because the experiences that are embodied in it have crystallized definitions as a reference that are socially built by legal, but also medical, religious, psychological and pedagogical systems [16,17]. They become "models" of what the family is and how it ought to be, grounded by memories that continue because they are kept alive in the groups and are continually maintained, since they are social memory frameworks that have a great strength in people's formation and belonging to groups [8].

Maybe for this reason it is possible to explain why the imprisoned elderly people, even when they are physically and/or emotionally distant, have a strong relationship with the family, whether carrying the positive or even negative memories. It is the family, the relationship center, and the blood ties that the elderly person maintains alive in their memory; they bring memories of the moments that determined their lives in the family, just as their sense of belonging to it.

As IE-10 states, "When it relates to family, it's a key factor to me. Everything begins with the family. My existence is my family". This statement of the elderly person leads us to think about the "family as something that is defined by a story that is told to the subjects, over time since they were born, by words, gestures, attitudes or silences, and that will be propagated by them and refocused in their own way" [18].

Despite not living with the family, or even not having a successful family relationship, the imprisoned elderly people carry the moments together with the family in their narratives—the memories with them—but the interviews recorded that feature with greater emotions, such as sorrow and crying.

The codification and counting of the most mentioned words were carried out, and afterward, the Word Cloud was generated (Figure 1). Among the most mentioned words, 10 were highlighted: family (48); children (21); mother (16); father (15); woman (13); life (13); crying (10); today (9); siblings (8); and longing (8).

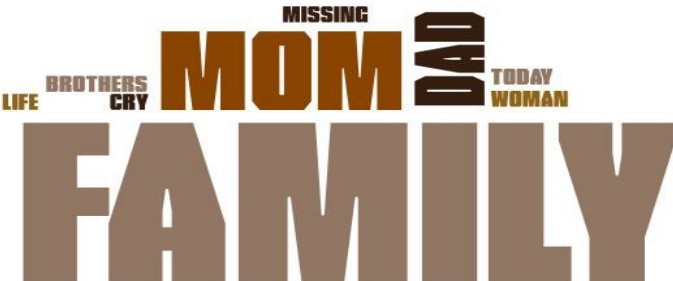

**Figure 1.** Word Cloud about family memories, Bahia, 2022.

By observing the words cloud in Figure 1 and the emphasis of the 10 most mentioned words, it is possible to suppose that the family brought up in the elderly people stories is comprised of those closest to them: children, mother, father, grandchildren, siblings and wife, sometimes labelled with the term "woman" by them. The characteristics of relationships, of homesickness and of regret for the way that the family was led are also

seen. For that matter, the assessments of this category around the closest family and the relations of the elderly people were carried out through the parts of the narratives that were tracked.

Thus, six categories emerged from the analysis: family as a reference point; missing the family; regrets; consequences of prison for the family; relations with father and mother; and family contradictions.

The analytical categories are presented below.

### 3.1. Family as a Reference Point

To remember, the subjects function as social frameworks, which are considered reference points. The family frames are one of those charts that are used as a field of significance in our lives [8].

> *Family is my foundation. The reason for my existence is my parents. They only have me as a child. (IE-09)*

> *To me, family is everything, without my family, I don't have anything, with my family, their support, we get lost, like I am here today. (IE-20)*

> *Family is all the best that I have in my life. Everything I have is my family. (IE-16)*

> *My family is everything. I have four daughters and nine grandchildren. When I talk about my family I cry, because my family is everything. (IE-02)*

> *Before prison, my life was good, I worked a lot, I was always very kind with my family, I always care of everyone around me too. My family is one of the best things I have in my life. (IE-14)*

### 3.2. Missing the Family

It is exactly because the elderly people have a sense of belonging in the family group that they carry in their reports the homesickness that sometimes makes them suffer and other times makes them strong to remain alive and look for a change in life.

> *Firstly, God in my life and, then, my family. I really miss my family, my grandchildren, everyone, I cry because of them. (IE-05)*

> *I miss my children, grandchildren and relatives. (IE-24)*

> *I miss my children and my wife the most. I really want to leave here to be with them. (IE-03)*

> *I miss everything, my family, my house, my grandchildren, sometimes I cry because of them. (IE-02)*

> *I am apart from my parents for over two years, and this hurts me a lot. The first thing I want to do when I leave here is to hug my parents. (IE-09)*

> *I really miss my children, my grandchildren, my daughters-in-law, I really wish to be with them, but they are the ones who give me strength to live, even with everything here. (IE-22)*

### 3.3. Regrets

Other points perceived in the participants' narratives were the regret of not having been a good husband, a good father, and the allegation of separation from his wife as having driven the conduct and, consequently, to the arrest.

> *Before splitting up from my wife, I was a hard-working man, I took care of the house and my family. I am divorced and lived alone, I drank a lot. My daughter and son are really important to me. My ex-wife is an amazing person, but I acted wrongly, I made a mistake, seriously wrong. I acted without thinking. (IE-11)*

> *I was always dedicated to my family and my job. My regret was to split up from my wife. (IE-31)*

*I really miss my son who lives with my mother-in law. He is the child of my second marriage, I really like him, I miss him, I suffer because of the distance, because I'm not able to see him. I wish to go back in time, when he was two, three, four-years-old, when we were close, Me, him and my wife who died. Being away from my son was the greatest punishment I had in my life. (IE-20)*

### 3.4. Consequences of Prison for the Family

Another occurrence seen in the narratives of the elderly was the consequences of imprisonment for the family. In the two following accounts, it is possible to perceive this fact.

*My wife is unsafe, people are stealing the groceries from my wife. My daughter takes my money, brings some to me, and the rest she buys groceries, but the relatives are stealing everything. (IE-23)*

*We see ourselves alone here, arrested, away from the family, and all the family was shaken. To be in prison is ridiculous, and no family is prepared for it. It's a defeat in the family, because there is the society, even us being poor. (IE-04)*

### 3.5. Relations with Father and Mother

Other memories noticed in the respondents' stories were the ones linked to the mother and father figures. They talked about the importance of the mother and their love for her. However, the memories concerning the fathers were related to cruelty, physical violence and the distancing from the school environment. Only in two speeches is the father displayed as important to the elderly people, as can be seen in the reports:

*My mother and father are older than 80-years-old, almost 90. I love my mother very much, I like my father, but it isn't the same, because he was too cruel with me, he used to spank me. (IE-17)*

*Firstly, I thank God for my life, and then my mother, she is everything in my life. (IE-10)*

*My father was very bad, but my mother was very loving. (IE-16)*

*My father beat me up. (IE-01)*

*My mother and my father are everything in my life. They are simple, they didn't have education access, but I love both of them very much. (IE-09)*

*My mother and father are my pride, they're from the countryside, they're pure, they aren't evil. They are my foundation. (IE-09)*

*My childhood was very difficult, my father compelled me to work. I begged him to let me study, but he didn't let me. Now with this age, 69-years-old, I can't learn anything more. I don't know how to read, nor write. (IE-23)*

*My father used to beat me up, but it wasn't for this that I didn't take care of him. When my mother used to talk about going to school, my father used to say that school was a hoe's handle. Today I am dumb, I don't know how to read anything nor write my name. (IE-31)*

### 3.6. Family Contradictions

In the following reports, another aspect that was revealed was the inconsistencies in the families, since the elderly people's stories characterized losses, ruptures, deaths, separation, sorrow and family breakdown.

*Everything in my life was hard . . . My mom was murdered by my father when I was a child . . . we saw everything. Shortly after, my father committed suicide. My siblings and I were all alone, without mother and without father, one taking care of the other. I got married young and, after a few years, I split up, I was without a wife. (IE-30)*

*I've already beaten my daughter up, with grass stem, I almost killed my daughter, and today she is the one that takes care of me, who visits me. I am sad about this, and sometimes I cry. (IE-16)*

*With the wife, I wasn't lucky. My marriage wasn't good ... I had many problems. I got divorced from my wife after a few years together. (IE-17)*

*Our family lived in peace. Our relation was always very good, but this uncontrolled situation happened. (IE-03)*

*I never got married, I am alone, I just have a few women in my life, but I didn't marry any of them, I just lived with one, but it didn't work out. She didn't like that I used to drink. I don't have children, I don't have anyone. (IE-26)*

## 4. Discussion

In spite of the social changes, crises and all the transitions that the family concept has been going through in recent years, it survives, adjusting itself to the most diverse forms, and is built and rebuilt. It is because there is no one in history who has lived distant from the notion of family; however, indeed, there are adjustments to different historical stages and socioeconomic and cultural situations [16].

In accordance with the elderly people's speeches, the family is the foundation, the reference point for their lives. Maurice Halbwachs, in his book *Les Cadres Sociaux de La Mémoire*, emphasized the family's role as the first constitution, the subject's reference point; only from a picture in this case, the family frame, it is possible rebuild the image of the people and the facts [8]. From this perspective, it is possible to state that the elderly participants of this study used facts and figures, as points of reference, to rebuild the family memories.

Therefore, in the context of family memory, there are many figures and facts that function as reference points, but each of these images indicates an entire profile, each of these facts summarize the whole period of the group's life; they are images and concepts. May our thinking be about them: everything probably happens as if we were reconnected to the past. However, this only means that, from the frame, we can rebuild images of people and facts [8]. By the comprehension of the aging phenomenon, as well as the family representations for these elderly people who nowadays live in an institution in the big city, what is realized is that these people value their family background, they have good memories of family life and appear to live in grief because of the lost bonds.

The elderly people mentioned the family as their life's basis, which may also be described by the fact that the human being has his first experiences as a society member in the family [19]; moreover, it is in the family we dedicate the largest share of our lives, besides which, most of our thoughts are linked to family thoughts [8].

Furthermore, the family is the essential place for the assurance of survival, the affective input, and the full protection of the children and other members, independent of the family's structured manner [20].

Aside from homesickness, firmly expressed in the reports, the fact that they are away from the family produces suffering for the elderly people. However, it is the distancing from the family that makes them stronger and allows them to achieve a life change. A study carried out with prisoners of different ages from the city of Rio de Janeiro, exhibited a similar outcome, as for the imprisoned subjects, "the family continues to be an elemental reference and an indubitable stimulus. When they think about the future, the prisoners talk about this primary core as a kind of talisman that helps them handle the current difficulties" [21]. Other points perceived in the respondents' speeches were regret for not being a good husband, a good father, and the claim that splitting up from their wives prompted the behavior that led to their imprisonment.

From the speeches, it was realized that the image of the woman/wife was described as very important to them, as a support to the family's formation, but especially for the maintenance of the ethical and moral values of the man's conduct required by society. This

view is related to the fact that the definition of family as nuclear has a secured space in Western culture, in which women occupy an essential role in motherhood and arise as a significant aggregator component for the survival of the family [22].

However, it is important to note that, although this discussion is not the focus of this study, it is believed that "the sacralization of the mother image comes as a way of repressing women's power and autonomy, from the establishment of a speech that will blame and threaten her if she does not comply with her natural and spontaneous maternal power" [22].

The retrospective perspective of the family life is dependent on the subjects' place in the family [8]. Maybe because most of the elderly were separated, their memories were primarily linked to their married life with their ex-wife, to the image of their children and to the regret for their behavior.

The consequence described by IE-23 is linked to the economic issue; and IE-04 describes the emotional suffering and shame. However, studies have shown, besides these, other significant effects on the lives of the relatives imprisoned, such as stigma, multiple roles and loss of support networks [23,24].

From this understanding, it can be seen that the practical consequences go beyond the prison walls, as they enter the homes of families and impact their entire structure and operation, even leading to social distancing and weaknesses.

In the period in which most of the elderly participants of this study were children, "the parental power was presumed to be a power relation of the father over his children, who were represented and raised without being recognized with rights in family relations" [25]. For this purpose, the physical violence, thus interpreted nowadays, was taken as natural in this period and formed a part of the paternal power. This reality remains in society, in spite of the changes that have happened, since violence and physical punishment as a manner of "educating" the child are still tolerated by our culture. Maybe this illustrates the reason why the elderly people remember the father image with a sense of sadness.

Regarding the mother figure, the stories are filled with nostalgia and positive memories, which are attached to moments lived next to that person who represented only love, dedication and support. It is believed that maybe this was not a reality experienced by all the elderly people, but the ones who carried memories of interaction with the family have in the mother's representation a kind person who they are proud to have.

Maurice Halbwachs [8] affirmed that the child's feelings, the ones he/she builds in the family with such intensity, do not comprehend the family's relational nature; in other words, for him, these feelings are built by simple blood relationships, which can only come from the family and be explained by it, so they cannot be defined by the mother's care, by the physical ancestry of the father, or by the usual living with brothers and sisters. Therefore, with this view, all feelings and attitudes are integrated or taught by the subjects in the family.

At first, the child embraces an attitude towards the father, the mother and the entire family, which is not just explained by the life intimacy, age difference, by the regular feelings of affection to those surrounding them, by respecting the creatures stronger than ourselves and on who we are dependent and by the gratitude for the services they provide to us. Such feelings, as intelligent as they may be, follow anticipated paths that do not rely on us, but which the society is careful to arrest. Actually, there is nothing less natural than this kind of affective display, nothing that is consistent with the principles and results more in this type of training [8].

Nevertheless, it is realized that, throughout life, people build feelings and conceptions which are shaped in the family and in their interaction groups and they rebuild their memories based on memories of the past in the face of the current moment. Therefore, it is supposed that, when remembering the mother and father figures, the perceptions are more linked to the present in which these subjects are placed.

What is exposed in the speeches is a debility in family bonds, just as the existence of violence, two issues that together or even individually impact the subjects' lives. Regarding

this, studies have demonstrated that living in familiar environments in which violence prevails and where the bonds are weak increases the odds that the subjects become victims of violence, but primarily use it, which may be a cultural matter which is naturalized in the family environment [26,27].

This way, the memories' analysis displayed the presence of weaknesses in the family relationships and violence in the lives of the elderly people. It also indicates that these elements are intrinsic to the difficulties in the process of building a family identity, just as the vulnerabilities that they are exposed to; the vulnerabilities are here taken as "unemployment risk, work precarity, poverty, lack of social protection or access to public services, the weakness of family and social bonds" [27], but even the "weakening of emotional bonds both relational and of belonging, also arising from age, ethnic, gender or disability discrimination, among others" [27].

It can be deduced, then, that vulnerable relationships, which may result in vulnerable families, wreck or prevent ties of social and family interaction, causing abandonment, lack of care and interpersonal bonds.

Faced with this context, it is possible to say that the subject joins a family through birth, marriage or by another way, but finds himself/herself in a group, which is shaped not only by personal feelings, but by rules and customs that do not depend of us, as they existed before us. Therefore, as we enter and/or build a family, we belong to their conducts, their choices and social situations, which are sometimes forced [8].

The family is important for the subjects' formation because it deals with the formation of personality and affects the individual's behavior through educational practices [28]. In addition, it is essential for the protection, growth and development of the core family. This complements the idea that "The family offers affective support and, mainly necessary materials for the development and welfare of its parts. It performs a major role in formal and informal education; it is in their space that ethical and moral values are presented and integrated, and it is where the ties of solidarity are reinforced" [27]; however, what is seen is the absence of public policies from the State's part and a decrease in social interventions.

On the basis of the above considerations, it is understandable that being in a family, whatever its setting, is not a choice. Therefore, the fragility of the bonds, the violence and the situation of social vulnerability can be something inherent to particular families, which requires fulfillment with social protection legislation; however, what is noticed is the absence of public support policies on the part of the State and a decrease in social interventions. In contrast, "it sets on the family an overload that it might not endure having in mind the situation of socioeconomic vulnerability" [20].

Generally, the elderly participants in the study described the family in a positive light and, in spite of being in a confinement situation and having memories about the family institution, which were sometimes contradictory, and with the fragility of the bonds, the family is still the balance mark in the life of the imprisoned elderly. This way, it is implied that the prison conserves the desire and the search for the family grouping.

## 5. Final Considerations

It was observed that the family formation has remained because of the experiences that are implied in it having socially crystallized definitions as a reference, rooted by memories' social frameworks that continue, once they remain alive in the groups and are permanently kept. It was also shown that, though the elderly people are not living with the family, or even the ones who throughout their lives did not have a close relationship with their family, they remembered the moments with the family with absolute affectivity and illustrate that it is the base of their lives. It was also noticed that the elderly people feel longing for the family and suffer because of the distance, but that is often the reason why they try to keep themselves strong and survive in the middle of such pressure that the prison environment imposes.

The outcomes of these stories reveal the mother figure as the centerpiece in their lives, and the parents' memories are linked to cruelty, physical violence and the impossibility

to complete their studies. The memories also exposed the fragilities and contradictions in family bonds, just as the violence that was reported; however, even with these experiences, the elderly people maintain their family in their memories as something positive and as a balance marker in their lives.

The present study addressed the gap in the literature regarding the theme of family and the incarcerated elderly person through sampling, providing interesting information about family relationships throughout life, but especially how the family becomes a potential strengthener of hope and survival in the prison environment.

The problems evidenced here are fundamental to show the need for effective interventions and support to strategies of greater insertion of the family in the prison context, in order to contribute to future public health interventions, aiming at a better awareness of prison leaders, as well as health professionals and the entire multidisciplinary team for the development of motivational work with the family to increase the support network for the incarcerated elderly.

One of the limitations was the size of the sample, so we emphasize the importance of studies like this one also at a national and international level, in order to continue the development of knowledge in the area and consequently the possibility of even more effective contributions to learning.

**Author Contributions:** Conceptualization, P.L., L.R. (Luana Reis) and L.R. (Luciana Reis); methodology, A.O., P.L., E.S. and A.L.; software, T.N. and A.O.; formal analysis, A.O., L.R. (Luciana Reis) and L.R. (Luana Reis); writing—preparation of the original draft, A.O., E.S. and P.L.; writing—proofreading and editing, A.O., L.R. (Luciana Reis), T.N. and L.R. (Luana Reis); view, L.R. (Luciana Reis) and A.L.; supervision, L.R. (Luciana Reis) and L.R. (Luana Reis). All authors have read and agreed to the published version of the manuscript.

**Funding:** This research received no external funding.

**Institutional Review Board Statement:** The study was conducted in accordance with the Declaration of Helsinki and approved by the Research Ethics Committee of the Independent Northeastern College (protocol code: No. 1.968.281).

**Informed Consent Statement:** Informed consent was obtained from all subjects involved in the study.

**Data Availability Statement:** The data sets of this study are available from the corresponding author upon reasonable request, contactable through: luciana.araujo@uesb.edu.br.

**Acknowledgments:** The authors would like to thank the Post-Graduate Program in Memory: Language and Society, Southwestern Bahia State University, Independent School of the Northeast, Brazil, and the participants who gave their time and shared their health experiences.

**Conflicts of Interest:** The authors declare no conflict of interest.

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
