# Peer review of "Incarcerated Brazilian Elderly: Memories about Family"

_2673-9259, doi:10.3390/jal2040024_

Round 1

Reviewer 1 Report

The paper addresses a significant issue, Memories of elderly incarcerated about family. The research, however, refers right from the title to the possibility of making only a qualitative contribution from a scientific point of view. The title is clear and in keeping with the content of the paper. What should, however, be deepened in the title is the clear reference of the study limited to the territory of Brazil alone as stated in the first line of the introduction.

The abstract is poorly structured in that it should not state its sessions with its name, which should be implied and discursive within it (e.g. 'Background', 'Methods' or 'Results'.

The introduction only refers to the Brazilian context and this should be justified since it is a problem of the memories of incarcerated elderly people about family issue that has affected all world.

The authors refer to exploratory and analytical study, with a qualitative approach, carried out in three prison units in Bahia, with 31 elderly people incarcerated, through semi-structured interview and their feelings. This has no scientific value to support the claim they are making that they should demonstrate the scientific reasons for choosing those  three prison units in Bahia, who experienced the same circumstance. Furthermore, the authors should declare in which project context this research is carried out and with which funding (e.g. national, European, trans-European funds, etc.). Furthermore, the introduction should end with a brief description of the contents that the reader will encounter in the paper according to the relevant section.

The methodology does not refer to a scientific method and this reduces a university approach to a journalistic one. The authors should specify how the sample of questions was constructed and why it is aimed at that type of stakeholder. Furthermore, a sample of 31 older people is not at all meaningful to defend any scientific data, all the more so if it is a pandemic phenomenon. The authors refer to a 'diary-file' that should be better described. What makes the data even less reliable, apart from the small number of respondents, is the period of administration of the questionnaire, which is only one month.

The session on the results is clear but it should be specified how the authors traced those factors and why they chose to categorise precisely those specific aspects. Annexes of what was obtained should be constructed for exhaustive collection. 

The results and discussion sections must appear separately, including only the information corresponding to each section 

The limitations and potential of the research should be included in the conclusion. The latter appear to be redundant and insufficiently structured.

Author Response

Dear reviewer,

Attached is the revised article, with the highlighted changes and the response letter.

Yours sincerely,

Reviewer 2 Report

Dears Authors 

Very interesting article. I just have a few points: 

1. the results section should be separated from the discussion section

2. use (-) and a nicknames in dialogs

3. enrich the conclusions section with solutions to the specific topic, interventions and research on the development of knowledge in this area

Good Luck!

Author Response

(The authors gave the same response as above.)

Reviewer 3 Report

The article undeniably contributes to research on people's attitudes toward their relatives. I highly appreciate (it is fascinating) that the survey covers the group of elderly people and people who are in prisons.

The methods used are correct and adequate for testing a small group (31 people).

However, I have a problem figuring out what the purpose of this study was. What does this research contribute to learning, and what actions can be implemented from it? Without extending the text with such application elements or better justifying the theoretical nature of the research, the text is not of research value.

I also miss an in-depth discussion with other studies.

Author Response

(The authors gave the same response as above.)

Round 2

Reviewer 1 Report

The authors have addressed all my comments and questions from previous report.